# Collaborative Gym: A Framework for Enabling and Evaluating Human-Agent Collaboration

**Yijia Shao[1], Vinay Samuel[2]\*, Yucheng Jiang[1]\*, John Yang[1], and Diyi Yang[1]**
[1]Computer Science Department, Stanford University
[2]Carnegie Mellon University
{shaoyj, diyiy}@cs.stanford.edu

## Abstract

While the advancement of large language models has spurred the development of AI agents to automate tasks, numerous use cases inherently require agents to collaborate with humans due to humans' latent preferences, domain expertise, or the need for control. To facilitate the study of human-agent collaboration, we introduce Collaborative Gym (Co-Gym), an open framework for developing and evaluating collaborative agents that engage in bidirectional communication with humans while interacting with task environments. We describe how the framework enables the implementation of new task environments and coordination between humans and agents through a flexible, non-turn-taking interaction paradigm, along with an evaluation suite that assesses both collaboration outcomes and processes. Our framework provides both a simulated condition with a reliable user simulator and a real-world condition with an interactive web application. Initial benchmark experiments across three representative tasks—creating travel plans, writing related work sections, and analyzing tabular data—demonstrate the benefits of human-agent collaboration: The best-performing collaborative agents consistently outperform their fully autonomous counterparts in task performance, achieving win rates of 86% in Travel Planning, 74% in Tabular Analysis, and 66% in Related Work when evaluated by real users. Despite these improvements, our evaluation reveals persistent limitations in current language models and agents, with communication and situational awareness failures observed in 65% and 40% of cases in the real condition, respectively. Released under the permissive MIT license, Co-Gym supports the addition of new task environments and can be used to develop collaborative agent applications, while its evaluation suite enables assessment and improvement of collaborative agents.

**Code & Data:** `https://github.com/SALT-NLP/collaborative-gym`

## 1 Introduction

Artificial Intelligence has long aspired to develop machines that act as teammates rather than mere tools (Nass et al., 1996; Russell & Norvig, 1995; Seeber et al., 2020). Recent advances in language models (LMs) have sparked growing interest in LM agents, with most research focused on developing *fully* autonomous agents to automate tasks—ranging from web navigation (Deng et al., 2023; Zhou et al., 2024), personal assistance (Drouin et al., 2024b; Shao et al., 2024b) to coding (Yang et al., 2024) and scientific discovery (Huang et al., 2024a; Majumder et al., 2024). Yet, numerous use cases inherently require human involvement due to latent preferences, domain expertise, or the need for control, even when LM agents can handle much of the workload. Beyond practical necessity, effective human-agent collaboration has the potential to achieve greater task performance compared to either alone, given their complementary expertise. Unfortunately, despite its importance, the human role remains largely overlooked in current LM agent research.

---

*Equal Contribution

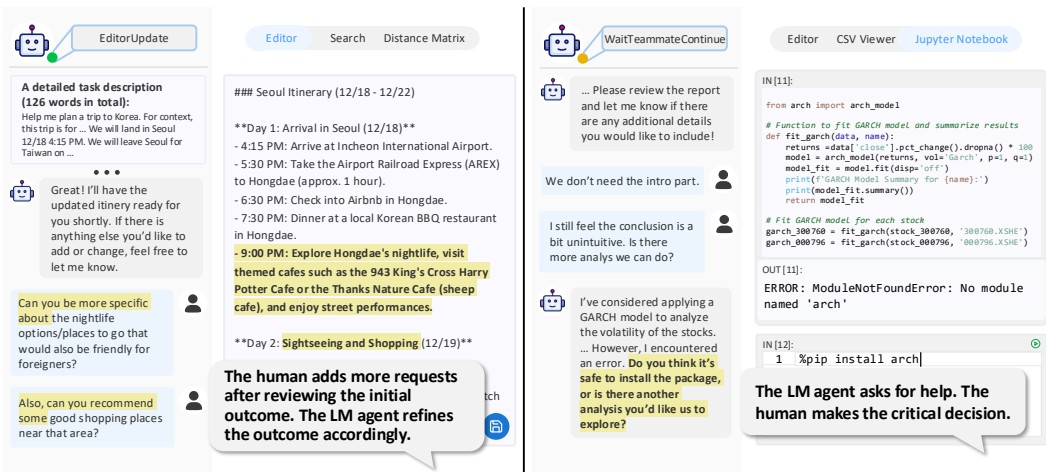

Figure 1: Collaborative Gym (Co-Gym) enables collaboration between humans and LM agents within a task environment. Left: Human adds requests and sends multiple messages without waiting for agent responses. Right: Human rates collaboration highly as the agent proactively seeks help when uncertain about package installation.

To address these gaps, we introduce **Collaborative Gym (Co-Gym)**, the first framework for developing and evaluating agents that can engage in bidirectional communication with humans while interacting with task environments. Co-Gym consists of three core components. First, *collaboration-driven environment design*. Co-Gym imposes no constraints on agent implementation but instead defines an environment interface that supports dual control in a shared workspace. Second, *a protocol for non-turn-taking interaction*. In Co-Gym, humans and agents coordinate through two collaboration acts and a notification protocol for real-time change monitoring rather than enforced turn structures (Skantze, 2021). Third, *an evaluation suite that assesses both outcomes and processes*. Co-Gym evaluates task delivery and performance while auditing initiative-taking patterns and human satisfaction during collaboration. We instantiate Co-Gym in both simulated conditions with a reliable LM-based user simulator and real conditions with an interactive web application featuring a chat panel and shared workspace. While more task environments have been added after the release of Co-Gym under the MIT license, in the paper, we support three representative tasks—creating travel plans, writing Related Work sections, and analyzing tabular data.

Co-Gym provides the infrastructure to answer the two fundamental questions: *(1) Is human-agent collaboration beneficial and in what ways? (2) How can we design LM agents that can collaborate with humans effectively?* We explore these questions by evaluating three types of agent architectures—fully autonomous agents that only interact with the environment, collaborative agents, and collaborative agents enhanced with a situational planning module—powered by four LMs under both simulated and real conditions. In the simulated condition where the agents interact with a LLM-based user simulator, our experiments show that compared to fully autonomous agents, collaborative agents have a lower delivery rate as human-agent teams sometimes fail to reach their goals within the step limit due to poor communication or coordination; but among the delivered cases, human-agent teams tend to achieve higher-quality outcomes. In the real condition, our results indicate that human-agent collaboration can be beneficial, with the best-performing collaborative agent achieving win rates of 86% in Travel Planning and 74% in Tabular Analysis compared to fully autonomous agents when evaluated by real users. Additionally, trajectories collected from Co-Gym and the evaluation suite reveal common failure modes in LM agents during collaborative processes, with communication and situational awareness emerging as the most prevalent issues. Our results underscore the need for advancements in both the underlying LMs and the agent scaffolding (*e.g.*, memory, tooling) to enable more effective and satisfying human-agent collaboration. Our main contributions can be summarized as follows:

- We introduce Collaborative Gym (Co-Gym), the first framework for enabling and evaluating human-agent collaboration with dual control over task environments and non-turn-

taking interaction. Co-Gym is released under the permissive MIT license to facilitate the study of collaborative agents.

- Experiments across three representative tasks, under both simulated and real conditions, highlight the benefits of human-agent collaboration. Co-Gym is publicly released to support the development of collaborative agent applications.
- Our evaluation and in-depth analysis reveal emergent dynamics in human-agent collaboration, along with weaknesses of current LM agents and underlying LMs.

## 2 RELATED WORK

**Infrastructure for Human-Agent Collaboration**

Despite increasing interest in developing LM agents, most work targets fully autonomous agents (Brockman et al., 2016), with limited research on infrastructure enabling human-agent collaboration. Existing approaches assume mostly human only single-control or dual-control with turn-taking. For example, WorkArena (Drouin et al., 2024a) and $\tau$-

Table 1: Comparison of Co-Gym with related work on enabling human-agent collaboration.

|  | **Control Mode** | **Coordination** | **Supported Env** |
|---|---|---|---|
| WorkArena | Single-control | N/A | Single (Browser) |
| CowPilot | Dual-control | Turn-taking | Single (Browser) |
| $\tau$-Bench | Single-control | N/A | Single (Database) |
| $\tau^2$-Bench | Dual-control | Turn-taking | Single (Database) |
| Co-Gym (Ours) | Dual-control | Non-turn-taking | Multiple |

Bench (Yao et al., 2024) are single-control settings whereas recent work such as, $\tau^2$-Bench (Barres et al., 2025) support dual-control. Notably however, these approaches still generally assume human-agent coordination follows a turn-taking structure.

The most closely related work is CowPilot (Huq et al., 2025), which introduces a dual-control, turn-taking framework for collaborative web navigation. At each step, users can pause, reject, or take alternative actions, providing a valuable testbed for studying human-agent coordination in navigation tasks. In contrast, Co-Gym relaxes the turn-taking constraint through a coordination protocol with collaboration acts and notifications, allowing humans and agents to act asynchronously when needed. Moreover, while CowPilot is specialized for web navigation, Co-Gym provides a general interface across diverse task environments, enabling broader investigations into when and how human-agent collaboration is beneficial (Vaccaro et al., 2024).

**Evaluation of LM Agents** Existing evaluations of LM agents often focus on end-to-end task success rates (Shridhar et al., 2021; Zhou et al., 2024; Xie et al., 2024b; Jimenez et al., 2024) or tool-use capabilities (Xu et al., 2023; Huang et al., 2024b). These evaluations typically involve single-step user instructions and binary task outcomes, without considering qualitative aspects of performance. While some studies have incorporated human-in-the-loop evaluation, they are generally limited to the simulated condition only (Yao et al., 2024; Zhang et al., 2024; Kim et al., 2022). Co-Gym addresses this limitation by supporting both simulated and real conditions within a unified framework, and includes an evaluation suite that assesses both collaboration outcomes and processes.

## 3 COLLABORATIVE GYM

We present Collaborative Gym (Co-Gym), a framework for enabling and evaluating human-agent collaboration. The design of Co-Gym comprises three core components: (1) collaboration-driven environment design (§3.1), (2) a protocol for non-turn-taking interaction (§3.2), and (3) evaluation of both outcomes and processes (§3.3). Figure 2 provides an overview.

### 3.1 TASK ENVIRONMENT

Within Co-Gym, we define each task as a Partially Observable Markov Decision Processes (POMDP) $(\mathcal{S}, \mathcal{A}, \mathcal{T}, \mathcal{R}, \mathcal{U}, \mathcal{O})$ with state space $\mathcal{S}$, action space $\mathcal{A}$, transition function $\mathcal{T} : \mathcal{S} \times \mathcal{A} \rightarrow \mathcal{S}$, reward function $\mathcal{R} : \mathcal{S} \times \mathcal{A} \rightarrow \mathbb{R}$, instruction space $\mathcal{U}$, and an observation space $\mathcal{O}$. Adding a new task environment (CoEnv) into Co-Gym requires specifying the tools available in $\mathcal{S}$, the corresponding $\mathcal{A}, \mathcal{O}$, and $\mathcal{T}$, and an initial task description as the instruction. In addition to the initial query, $\mathcal{U}$ also includes instructions that emerge during the collaboration process.

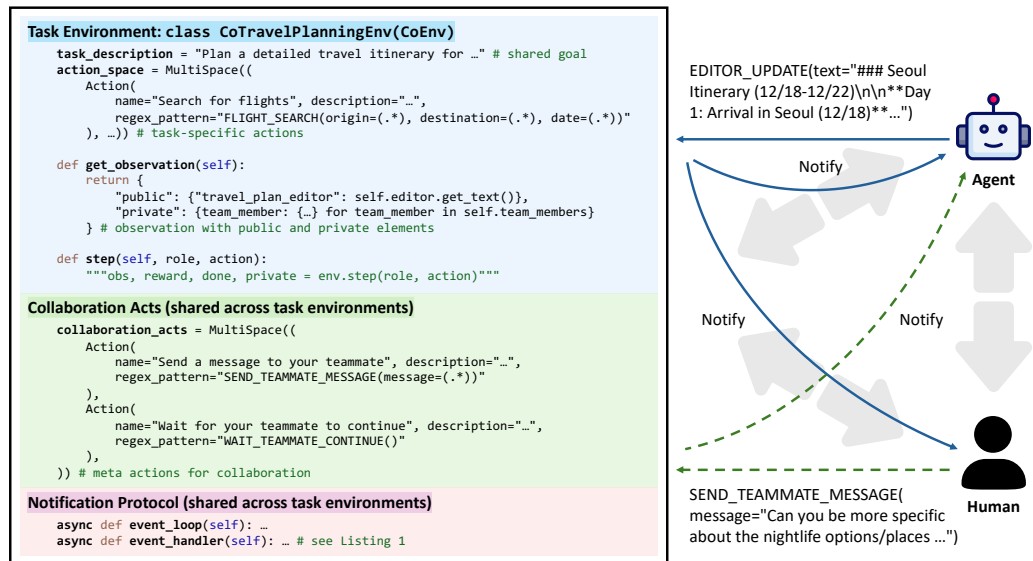

Figure 2: Overview of Co-Gym framework. The task environment interface (`CoEnv`) requires specifying the task description, action space, and observation space (§3.1). The collaboration acts and notification protocol (§3.2) are shared across tasks. For example, when the agent updates the public component, both parties are notified with the new observation (blue solid lines); parties can coordinate by sending messages (green dashed lines).

To support actions from both humans and agents within a shared task environment, `CoEnv` introduces a `role` parameter in its `step` function, which allows the environment to be updated based on the role-specific `action`. Moreover, even within a shared environment, the observation space can include both public and private components, analogous to human teams where some components (*e.g.*, whiteboards) are shared while others (*e.g.*, personal notebooks) remain private. `CoEnv` allows such flexibility by supporting differentiated observations for different parties using a `private` flag to distinguish between actions affecting shared components versus private components of the action taker. Thus, the resulting `CoEnv` abstraction is: `obs, reward, done, private = env.step(role, action)`

## 3.2 COORDINATION

Co-Gym removes turn-taking restrictions by coordinating via the collaboration acts and a notification protocol that are shared across task environments.

**Collaboration Acts** To mirror natural human collaboration, where coordination occurs through effective communication, Co-Gym augments the task-specific action space with two meta-actions: `SendTeammateMessage` for exchanging messages between teammates, and `WaitTeammateContinue` which serves as a keep-alive signal. Agents can proactively trigger `SendTeammateMessage` without requiring prior human messages. Each party can also send multiple consecutive messages without waiting for responses (see Figure 1 for example).

**Notification Protocol** While humans can continuously monitor their environment, agents require programmatic notification of changes. Co-Gym adopts the following notification protocol that operates on four event types: (1) shared observation updates, which broadcast notifications to all parties; (2) private observation changes, which notify only the associated party; (3) new messages, which trigger notifications for all recipients; and (4) environment inactivity exceeding a specified temporal threshold, which broadcast notifications to all parties. For example, as illustrated in Figure 2, when the agent updates the public component (*i.e.*, Editor), both parties are notified with the new observation (blue solid lines); when the human sends a message, only the agent is notified with the new observation (green dashed lines). We use a Redis server to manage notifications across different processes and Listing 1 provides the pseudo code of this notification protocol.

### 3.3 EVALUATION SUITE

Existing evaluations of LM agents often target task success rates. While task completion is crucial, the outcome quality and collaboration process also play a significant part. Co-Gym framework provides an evaluation suite that assesses both collaboration outcomes and processes.

**Evaluating Collaboration Outcome**   We assess the collaboration outcome along two dimensions:

- DELIVERY RATE: This binary metric indicates whether collaborative agents can successfully deliver a task outcome within a predefined step limit.

- TASK PERFORMANCE: A task-specific scoring function evaluates the quality of the outcome for delivered cases. This function may be a deterministic metric or based on LM/human judgments. Scores are normalized to the range $[0, 1]$ for comparability across tasks.

**Auditing Collaboration Process**   Co-Gym framework includes two process auditing metrics:

- INITIATIVE-TAKING: Collaborative agents operate as mixed-initiative systems (Horvitz, 1999). To audit how the initiative is shared in human-agent teams, we introduce *Initiative Entropy* ($H_{\text{init}}$), which quantifies the balance of initiative among team members. Building on Chu-Carroll & Brown (1997), we define an utterance in collaborative dialogues as *exhibiting initiative* if it directs task execution or establishes mutual belief, and employ an LM to annotate utterances (see Figure 12 for the judging prompt and examples). Using entropy as a measure assigns higher values to more uniform distributions and lower values to more skewed ones. Specifically,

$$H_{\text{init}} = \begin{cases} -\Sigma_{i=1}^{N} p_i \log_N(p_i) & \forall i, p_i > 0, \\ 0 & \exists i, p_i = 0 \end{cases} \quad (1)$$

  Here, $p_i$ is the proportion of initiative-taking utterances by party $i$ and $N$ refers to the total number of parties.

- OVERALL SATISFACTION: At the end of the teamwork, we collect human ratings of their collaboration experience with the agent using a 1–5 Likert scale, complementing the evaluation of task performance.

## 4 CO-GYM INSTANTIATIONS

We instantiate the Co-Gym framework in both simulated and real-world conditions. In this paper, we evaluate three representative tasks that highlight distinct facets of human-agent collaboration: latent personalized preferences, domain knowledge requirements, and compatible expertise between humans and agents. Additional task environments (*e.g.*, computer-use environments (Xie et al., 2024b)) have been contributed since our open-source release.

### 4.1 SUPPORTED CONDITIONS

**Co-Gym (Simulated)**   We create a sandbox environment where human-agent collaboration can be studied without impacting real-world environments or requiring real human participants. Each task is associated with a set of pre-collected instances that define concrete shared goals for the human-agent team. Tools within each task environment are mocked using static databases to emulate realistic interactions in a reproducible setup.

Following the practice in previous work (Wu et al., 2025; Yao et al., 2024; Barres et al., 2025), we build an LM-based user simulator to simulate human behavior. To introduce dynamics typical of human-agent collaboration, where humans possess additional knowledge and preferences about the task, we provide the simulator LM with hidden information—pre-curated insights associated with each task instance. Figure 3 illustrates the user simulator workflow and we validate the simulator quality in Appendix D.

**Co-Gym (Real)**   While experiments with simulated humans provide a valuable surrogate for advancing human-agent collaboration, they cannot fully replace human evaluation. We instantiate Co-Gym (Real) as a web application, enabling users to easily perform the three supported tasks directly through their web browsers. Details of our user interface are provided in Appendix C.4. To

Table 2: Summary of action space, observation space, and data used in Co-Gym (Simulated). Implementation details for these environments are provided in Appendix C.2.

| Task | Action Space ($\mathcal{A}$) | Observation Space ($\mathcal{O}$) | | Data for Co-Gym (Simulated) | | |
|---|---|---|---|---|---|---|
| | | Component | Private? | Data Source | # Instances | Grader for Task Perf. |
| Travel Planning | `CitySearch`, `AttractionSearch`, `RestaurantSearch`, `FlightSearch`, `AccommodationSearch` `DistanceMatrix` `EditorUpdate` | Search window

Distance matrix
Editor | True

True
False | TravelPlanner (Xie et al., 2024a) Validation Subset | 102 | Grader from (Xie et al., 2024a) |
| Related Work | `SearchPaper` `LibraryAddPaper`, `LibraryDropPaper` `LibraryToDraft`, `EditorUpdate` | Search window
Paper library
Editor | True
False
False | Subset of arXiv CS Papers | 100 | Rubric Grading (Figure 13) |
| Tabular Analysis | `/` `JupyterExecuteCell` `EditorUpdate` | Tabular data
Jupyter notebook
Editor | False
False
False | DiscoveryBench Subset (Majumder et al., 2024) | 110 | Grader from (Majumder et al., 2024) |

incentivize humans in real-world evaluations, we instantiate the transition function within Co-Gym (Real) by leveraging real tools (*e.g.*, Google Maps, arXiv search) within the task environments.

## 4.2 SUPPORTED TASKS

**Travel Planning** Travel planning is a widely sought yet complex task, as optimal solutions depend on users' latent preferences and constraints that may not be explicit in the initial query. The Travel Planning environment in Co-Gym provides a comprehensive action space, including various search functions, aligning with Xie et al. (2024a) which studies fully autonomous agents. The search window is private in the observation space, while the task action space also includes `EditorUpdate` that modifies the travel plan visible to both the agent and human user.

**Related Work Writing** Grounded article writing is a task commonly used to assess agentic systems (Shao et al., 2024a; Wang et al., 2024) and users usually possess additional domain knowledge when they use agents to write the Related Work section. The Related Work environment supports `SearchPaper` action to retrieve papers based on the given query. A shared library and text editor are included in the observation space, enabling actions like `LibraryAddPaper`, `LibraryDropPaper`, `LibraryToDraft`, and `EditorUpdate`.

**Tabular Analysis** Another common application of LM agents is data-driven research, where researchers and agents can leverage different expertise: researchers contribute domain data while agents usually excel at coding and statistical analysis (Majumder et al., 2024). The Tabular Analysis environment includes a shared Jupyter Notebook and text editor, supporting actions such as `JupyterExecuteCell` and `EditorUpdate`. The goal of this task is to derive analytical insights from the provided data and initial query.

## 5 EXPERIMENT

The goal of the Co-Gym framework is to enable human-agent collaboration with dual control over task environments and non-turn-taking interaction. In this section, we conduct three sets of experiments: (1) **benchmark the performance of different agent types** to understand how collaborative agents compare with fully autonomous agents and how different models and agent designs affect performance; (2) **analyze human-agent collaboration patterns** using trajectories collected in Co-Gym; and (3) **ablate the notification protocol** to compare the Co-Gym interaction paradigm with the turn-taking paradigm common in chatbots.

## 5.1 BENCHMARKING DIFFERENT AGENT TYPES

We evaluate 3 agent types powered by 4 LMs: GPT-4o (`gpt-4o-2024-08-06`), GPT-4-turbo (`gpt-4-turbo-2024-04-09`), Claude-3.5-sonnet (`claude-3-5-sonnet-20241022`), Llama-3.1-70B. The agent types are as follows: (1) **Fully Autonomous Agent** which only interacts with the task environment. (2) **Collaborative Agent** which adopts the same implementation but extends the action space to include both task-specific actions and collaboration acts (*i.e.*, `SendTeammateMessage`, `WaitTeammateContinue`). (3) **Collaborative Agent with Situational Planning** which employs a two-stage decision-making approach when processing notifications. First, the LM makes a 3-way decision based on all available information (*i.e.*, task description,

Table 3: **Results in Co-Gym (Simulated).** The human is simulated by `gpt-4o`. * denotes significant improvement on Task Performance (Task Perf.) over the Fully Autonomous Agent powered by the same LM ($p < 0.05$; McNemar test for Tabular Analysis due to dichotomous task-specific scoring function and pairwise $t$-test for other tasks). For the auditing metric Initiative Entropy ($H_{\text{init}}$), higher numbers indicate that team members take initiatives more equally.

| | | Travel Planning | | | Related Work | | | Tabular Analysis | | |
|---|---|---|---|---|---|---|---|---|---|---|
| | | Delivery Rate | Task Perf. | $H_{\text{init}}$ | Delivery Rate | Task Perf. | $H_{\text{init}}$ | Delivery Rate | Task Perf. | $H_{\text{init}}$ |
| Fully Autonomous Agent | GPT-4o | 0.873 | 0.591 | / | 0.960 | 0.583 | / | 0.991 | 0.408 | / |
| | GPT-4-turbo | 0.980 | 0.615 | / | 0.940 | 0.575 | / | **1.00** | 0.426 | / |
| | Claude-3.5-sonnet | **0.990** | 0.577 | / | 0.970 | 0.617 | / | **1.00** | 0.358 | / |
| | Llama-3.1-70B | 0.745 | 0.646 | / | 0.960 | 0.727 | / | 0.836 | 0.358 | / |
| Collaborative Agent | GPT-4o | 0.745 | 0.641 | 0.42 | 0.980 | 0.588 | 0.16 | 0.927 | 0.311 | 0.10 |
| | GPT-4-turbo | 0.931 | 0.642 | 0.05 | 0.930 | 0.628 | 0.00 | 0.900 | 0.351 | 0.06 |
| | Claude-3.5-sonnet | 0.677 | 0.653* | 0.48 | 0.950 | 0.621 | 0.04 | 0.891 | 0.359 | 0.02 |
| | Llama-3.1-70B | 0.706 | 0.703* | 0.28 | 0.930 | 0.675 | 0.10 | 0.746 | 0.427 | 0.23 |
| Collaborative Agent with Situational Planning | GPT-4o | 0.735 | 0.667* | 0.90 | 0.970 | 0.658* | 0.79 | 0.891 | **0.434*** | 0.40 |
| | GPT-4-turbo | 0.853 | 0.703* | 0.43 | 0.950 | 0.604 | 0.27 | 0.846 | 0.428 | 0.09 |
| | Claude-3.5-sonnet | 0.941 | 0.682* | 0.80 | 0.900 | **0.736*** | 0.55 | 0.946 | 0.365* | 0.74 |
| | Llama-3.1-70B | 0.706 | **0.707*** | 0.70 | **0.990** | 0.679 | 0.70 | 0.736 | 0.402 | 0.62 |

chat history, action history, observations) to take a task action, send a message, or do nothing. If it chooses to do nothing, the next action would be `WaitTeammateContinue`; otherwise, it is further prompted to generate the final action string. Figure 5 illustrates the workflow of this agent. All agent types are implemented with ReAct (Yao et al., 2022) which requires the LM to output "thought" before generating the action. We incorporate a Scratchpad module as an in-session memory for all agents (Sumers et al., 2023). This memory is updated by the same LM before determining the next action. All LMs are used with a temperature of 0.

Table 3 and Table 4 summarize results under simulated and real conditions respectively.

### 5.1.1 RESULTS IN CO-GYM (SIMULATED)

**Under the same step limit, collaborative agents struggle more to complete the task.** When we simulate human users using an LLM-based user simulator, human-agent collaboration exhibits a lower Delivery Rate compared to Fully Autonomous Agents in the simulated condition with a step limit of 30. This could be attributed to the inherent complexity of decision-making for collaborative agents which must adapt plans frequently based on human actions or messages. Analysis of concrete cases (§5.2) revealed that failures stemmed mainly from the agent ignoring human messages (C.2, 46%), prompting repeated messages (SA.2, 26%), or repeating and omitting actions (PL.2, 33%).

**Among completed tasks, collaborative agents lead to better task performance.** Collaborative Agent with Situational Planning achieves the best Task Performance across all three tasks and has significant improvement over the Fully Autonomous Agent powered by the same LM. Notably, the agent powered by Claude-3.5-sonnet achieves a 0.105 improvement in Travel Planning (maximum score: 1) and a 0.119 improvement in Related Work. Compared to Fully Autonomous Agents, collaborative agents engage in a more iterative process. For example, before starting to plan the trip based on the initial query, the agent might ask, "Are there any particular cities or attractions you want to visit?"; after drafting the first version of a related work section, the simulated human might suggest, "Could you please add headings?". This iterative interaction improves task outcomes.

### 5.1.2 RESULTS IN CO-GYM (REAL)

We recruited human participants with relevant expertise or practical needs to collaborate with Collaborative Agent with Situational Planning powered by `gpt-4o` in Co-Gym (Real). In total, 99 unique individuals participated in the study, contributing 150 human-agent collaboration trajectories. These trajectories consist of 6.3k actions performed by human-agent teams and over 77k words of verbal communication. We ask humans to rate the final outcome on a 1-5 scale (1: "Extremely dissatisfied", 2: "Somewhat dissatisfied", 3: "Neutral", 4: "Somewhat satisfied", 5: "Extremely satisfied"). All Task Performance scores are normalized to the range $[0, 1]$ according to §3.3. Appendix F.3 includes more human evaluation details.

**Collaborative agents lead to higher Task Performance and Overall Satisfaction.** Consistent with the simulated condition, collaborative agents outperform autonomous agents across all tasks in the real condition in terms of Task Performance. The Delivery Rate reaches 100% because human participants are proactive in ensuring task completion. However, Overall Satisfaction with the collaboration process varies by task. In Tabular Analysis, which achieves the highest Overall Satisfaction of 4.06, the agent handles

Table 4: **Results in Co-Gym (Real).** The win rate is calculated against the outcome produced by the Fully Autonomous Agent, using the provided query as the task description. The sample size for each task is 50.

| | Travel Planning | Related Work | Tabular Analysis |
|---|---|---|---|
| Delivery Rate | 1.00 | 1.00 | 1.00 |
| Task Performance (Automatic Rating) | / | 0.660 | / |
| Task Performance (Human Rating) | 0.788 | 0.604 | 0.804 |
| Win Rate vs. Autonomous Agent | 86% | 66% | 74% |
| 95% Confidence Interval | [0.764, 0.956] | [0.529, 0.791] | [0.618, 0.862] |
| Initiative Entropy ($H_{init}$) | 0.88 | 0.63 | 0.74 |
| Overall Satisfaction | 3.78 | 3.06 | 4.06 |
| UserEnvActRatio | 0.24 | 0.32 | 0.21 |

most of the work in writing, executing, and interpreting code, while humans primarily contribute by posing new questions for further analysis. In contrast, the Related Work task receives low satisfaction scores (3.06 out of 5 where 3 indicates "Neutral" on the Likert scale), as users often come with domain expertise and strong preferences regarding how relevant papers should be organized. Therefore, users frequently need to provide step-by-step instructions to guide the collaboration, as reflected in the low $H_{init}$ score, which indicates unbalanced initiative-taking. These findings align with the idea that task difficulty in human-AI collaboration depends not only on technical complexity but also on user expectations and expertise (Spitzer et al., 2023).

**Users voluntarily allocate 21-32% of their actions to direct environment control.** To assess whether dual control provides meaningful additional flexibility for human users, we further compute UserEnvActRatio, the proportion of user actions taken directly on the task environment relative to all user actions which also include the collaboration acts. This metric quantifies how often users choose to intervene in the environment rather than only communicate with the agent, thereby indicating the extent to which users actually leverage dual control. Across the three tasks studied in this work, users voluntarily allocate 21-32% of their actions to direct environment control, demonstrating that users actively value and exploit this complementary interaction channel. Notably, UserEnvActRatio is highest for the Related Work task, suggesting that users rely more on dual control when a task demands fine-grained edits or when they possess stronger domain knowledge that they wish to directly apply.

## 5.2 ANALYZING HUMAN-AGENT COLLABORATION TRAJECTORIES

**Co-Gym enables the discovery of effective patterns in human-agent collaboration.** Although fully autonomous agents may complete the task on its own, we observe several key components of successful human-agent collaboration by analyzing trajectories with high Task Performance. One common pattern in successful collaboration is *proactive communication*. In Figure 14, we demonstrate a case of successful collaboration where the user poses a broad subjective question regarding activity planning, and the agent responds with relevant suggestions, waiting for the user's approval before making changes. Another emerging pattern is the *distribution of work based on expertise*. For example (Figure 15), when collaborating on a related work section about "Software Techniques for Emerging Hardware Platforms," the LM agent asks the human to narrow the search range for embedding methods in NLP tasks, as it is tangential to the topic but previously mentioned. Throughout the collaboration, the LM agent primarily handles searching and writing, while the human focuses on reviewing the results and adjusting cited papers. Lastly, we observe the potential for *collaborative agents to enhance human control*. For instance, when allowed to communicate with humans or wait for their input, the LM agent asks the human to make critical decisions, such as whether to install a specific package.

**Error analysis reveals core challenges for LM agents when collaborating with humans and demonstrates similarities between simulated and real conditions.** We developed a failure mode annotation taxonomy through a single-pass review of all real human-agent collaboration trajectories collected from Co-Gym (Real). The resulting 26 error types were organized into five high-level categories: Communication (C), Situational Awareness (SA), Planning (PL), Environment Awareness (EA), and Personalization (P). Three authors independently hand-annotated 150 trajectories

Table 5: **Summary of error analysis.** Detailed second-level categories (e.g., C.1, C.2) are provided in Table 10, along with an in-depth discussion of error attribution to the underlying LM and agent scaffolding. The error distributions between Co-Gym (Real) and Co-Gym (Simulated) have a Spearman rank correlation of 0.803 ($p = 7.94 \times 10^{-7}$).

| Category | Description | Real | Simulated |
|---|---|---|---|
| Communication (C.1-C.7) | Failures in maintaining effective information exchange, that disrupt understanding, coordination, or task execution. | 65% | 80% |
| Situational Awareness (SA.1-SA.6) | Failures in contextual understanding and reasoning about the current state of the task or collaboration. | 40% | 47% |
| Planning (PL.1-PL.6) | Failures in devising, updating, or executing coherent plans, especially in dynamic or long-horizon scenarios. | 39% | 43% |
| Environment Awareness (EA.1-EA.4) | Failures in recognizing or accounting for operational constraints and resources within the task environment. | 28% | 13% |
| Personalization (P.1-P.3) | Failures in adapting behaviors to align with individual user in-session preferences and interaction patterns. | 16% | 11% |

each from Co-Gym (Real) and Co-Gym (Simulated) conditions based on the taxonomy. As shown in Table 5, we found that trajectories collected from these two conditions exhibit many similarities. The error distributions between the two conditions have a Spearman rank correlation of 0.803 ($p = 7.94 \times 10^{-7}$), indicating strong alignment.

The most prevalent issues involve Communication and Situational Awareness (Raiman et al., 2019). For instance, in Figure 16, the agent fails to update collaborators on its status (C.1 *"Agents process tasks without informing users, resulting in a lack of progress awareness"*, C.3 *"Agents do not provide progress updates or notify users upon task completion"*) and provides incorrect summaries after executing actions (C.5 *"Agents provide inadequate summaries after executing actions"*), causing human confusion and disrupting collaboration. In Figure 17, the agent fails to decide when to ask the human for help or to incorporate the human's suggestions, repeatedly encountering the same issues during code execution and becoming trapped in an endless loop (SA.1 *"Agents disregard session context, treating each request as an isolated task"*, SA.2 *"Repetitive queries arise from neglecting prior interactions and user feedback"*, SA.3 *"Agents fail to process multiple user messages cohesively"*). These errors highlight the limitations of current LMs when deployed in complex, agentic setups with human involvement. In Appendix G Table 10, we further attribute each second-level error category to the underlying LM and agent scaffolding.

## 5.3 ABLATION STUDY

Turn-based interaction, where human agent coordinate by taking their turns, is well-suited for conversational scenarios like chatbots. However, Co-Gym targets human-agent collaboration on real tasks (writing, analysis, *etc.*), where parallel progress and opportunistic human intervention are more desirable. To validate this design choice, we compare Co-Gym with a turn-taking version.

Table 6: **Ablation results of the notification protocol.** Turn-taking Co-Gym removes the notification protocol and requires humans and agents to coordinate through turn-taking. The sample size for each task is 20.

| | Travel Planning | | | Tabular Analysis | | |
|---|---|---|---|---|---|---|
| | Task Perf. | Overall Satisfaction | Win Rate | Task Perf. | Overall Satisfaction | Win Rate |
| Turn-taking Co-Gym | 0.81 | 3.55 | 30% | 0.79 | 3.55 | 30% |
| Co-Gym | 0.84 | 3.95 | 70% | 0.81 | 4.25 | 70% |

We implemented the turn-taking version by removing the notification protocol and constraining interaction so that humans and agents must alternate actions. In this version, `WaitTeammateContinue` can be used to skip the current turn, and we added an overlay on the web application that disables user actions during the agent's turn. We recruited 20 professional travel planners and data analysts through Upwork for Travel Planning and Tabular Analysis, respectively, to evaluate the Collaborative Agent with Situational Planning powered by `gpt-4o` in both versions. We also collected pairwise preferences between the two versions.

Table 6 presents the ablation results. In Travel Planning, Co-Gym achieves higher task performance (0.84 vs 0.81), greater overall satisfaction (3.95 vs 3.55, $p = 0.012$ in pairwise $t$-test), and a substantial preference advantage (70% win rate). A similar trend holds in Tabular Analysis. Notably, the agent implementation remains identical across both versions—only the interaction paradigm differs. Qualitative feedback revealed that participants found the non-turn-taking interaction more efficient and teammate-like, as participants mentioned, "Non–turn-taking felt more natural and flexible: the agent didn't wait for every instruction and started working like a real teammate; I could step in anytime to guide it", "Output quality was good in both; non–turn-taking was effective. A remaining challenge is that the system should ask for help when it hits an error—ideally it can do both simultaneously."

## 6  CONCLUSION

We introduce Collaborative Gym (Co-Gym), the first framework designed to evaluate and facilitate human-agent collaboration with dual control over task environments and non-turn-taking interaction. By providing an evaluation suite that assesses both collaboration outcomes and processes, Co-Gym enables comprehensive benchmark experiments that demonstrate the potential for building collaborative agents. Our results show that human-agent teams achieve superior performance compared to fully autonomous agents, while simultaneously exposing critical challenges in communication, situational awareness, and planning that necessitate advancements in both underlying LMs and agent design. Released under the MIT license, Co-Gym holds the potential to facilitate the study of human-agent collaboration and advances the broader goal of creating AI systems that augment human capabilities rather than replace them.

## ETHICS STATEMENT

**Human Study Ethics**   This research involves human participants in the Co-Gym (Real) condition, and all human studies were conducted under IRB approval to ensure participant safety and data protection.

**Broader Impact**   The development of collaborative agents has the potential to transform human-computer interaction across industries. From enhancing productivity in knowledge work to improving safety in high-stakes domains, collaborative agents could augment human capabilities in meaningful ways. However, there are risks associated with this technology. Miscommunication or over-reliance on agents could lead to errors in critical tasks, particularly if agents' limitations in communication or situational awareness are not mitigated. Furthermore, suppose LM agents become increasingly integrated into human workflows, ethical concerns related to bias, privacy, and accountability must be addressed. Ensuring that these systems are designed to respect human autonomy, uphold fairness, and transparently communicate their limitations will be critical. By focusing on human-agent collaboration rather than full automation, Co-Gym aligns with the broader goal of creating AI systems that value human agency and preserve human control.

## REPRODUCIBILITY STATEMENT

To ensure reproducibility of our results, we provide comprehensive documentation and materials across multiple components. The complete Co-Gym framework, including all task environments, agent implementations, and evaluation scripts, is available as open-source code under the MIT license, with the anonymized repository submitted as supplementary materials for review. All experimental details necessary to reproduce our results are specified in §5.1, including the evaluated language models, hyperparameters (temperature settings, step limits), and agent architectures. Implementation details of task environments are thoroughly documented in §4.2, while human evaluation protocols, participant compensation details, and IRB approval information are provided in Appendix F.3. Statistical significance testing methods and error bar calculations are detailed alongside our experimental results in the main text. The interactive web application interface is illustrated in Figure 4 with screenshots, and all code includes sufficient documentation and setup instructions to enable faithful reproduction of our benchmark experiments across both simulated and real conditions.

ACKNOWLEDGMENTS

This work was supported by Open Philanthropy, Schmidt Sciences, and a grant under the NSF CAREER IIS-2247357, ONR N00014-23-1-2420 and ONR N00014-24-1-2532. Claude credits are partially funded through the Anthropic External Researcher Access Program. We are thankful to Yanzhe Zhang, Chenglei Si, Hao Zhu, William Held, Ryan Louie, Omar Shaikh, Shannon Shen, Zora (Zhiruo) Wang, Eric Zelikman, and Ioana Ciucă for their helpful suggestions and feedback at different stages of this project.

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

# Appendix

## Table of Contents

## A  LIMITATIONS & FUTURE DIRECTIONS

Despite its contributions, Co-Gym has certain limitations. First, our study focuses on three tasks that, while representative, do not encompass the full spectrum of human-agent collaboration scenarios. Including a wider variety of tasks, such as creative design or real-time decision-making, could provide more comprehensive evaluation of collaborative agents. Notably, the unified environment interface in Co-Gym facilitates the addition of new task environments, and additional environments (*e.g.*, computer-use tasks) have been contributed since our open-source release. Second, the framework relies primarily on simulated conditions for agent comparison. While we validated the user simulator quality in Appendix D and our trajectory analysis demonstrates similar error patterns between Co-Gym (Real) and Co-Gym (Simulated), exploring how to leverage simulated conditions to improve collaborative agents remains a meaningful direction for future work. Finally, our real-world experiments, though valuable and sufficiently powered for statistical significance, involve a relatively limited number of human participants. Deploying Co-Gym (Real) more broadly to include diverse user populations across varying expertise levels represents a critical next step for validating our findings at scale.

Finally, in terms of the scope of our contributions, Co-Gym centers on dual-control, non-turn-taking interaction. We do not claim that all collaborative tasks should adopt this paradigm. For example, chatbots would still be valuable for tasks that can be effectively completed through standard turn-taking conversations. Dual-control, non-turn-taking interaction is most beneficial in settings where (1) collaboration unfolds within a shared, manipulable workspace, and (2) users have incentives to directly shape the involving artifact or have needs for fine-grained control. Even within this subset, the optimal interaction setup is inherently task-dependent. Our goal is not to prescribe that all collaborative systems adopt this style, but to provide a principled framework for studying collaboration when such interaction is desirable. To this end, Co-Gym offers a reusable scaffold: researchers can plug in their own environments and empirically assess whether dual control and non–turn-taking dynamics benefit their particular task. We hope the community will collectively build understanding about when this mode of interaction is most effective for human-agent collaboration.

## B  DISCLOSURE OF LLM USAGE

Large language models were used solely to assist with writing and polishing the manuscript text, but were not involved in research ideation, methodology development, or information retrieval. Additionally, we utilized a collaborative agent within our own Co-Gym Tabular Analysis task environment to assist with statistical significance testing and analysis of experimental results. All results have been verified by the authors.

## C  IMPLEMENTATION DETAILS OF CO-GYM

### C.1  INFRASTRUCTURE & API

As introduced in §3, Co-Gym defines an API for task environments that enables multiple parties to take actions in a shared workspace. The API requires a `role` parameter in the `step` function and additionally returns a `private` flag to indicate whether a change needs to notify all parties. Leveraging the additional flag, Co-Gym further establishes a notification protocol to coordinate non-turn-taking interaction, eliminating the rigidity of turn-based interaction. This protocol is implemented using a Redis server, which facilitates communication across different components (*i.e.*, the environment, agent, or simulated user in Co-Gym (Simulated) condition) to send and listen to messages on designated channels. Specifically, the environment sends notifications to each party (*i.e.*, `role`) through the {role}/obs channel and listens for updates on the `step` channel. Listing 1 provides the pseudo code of the `event_handler` implementing this notification protocol.

Notably, *Co-Gym is not an agent framework* but a framework designed to apply and evaluate LM agents in various task environments alongside human collaborators. Our design principles consider an LM agent as an LM-empowered system comprising the underlying LM(s), prompts, and additional scaffolding (*e.g.*, memory, external tools, *etc.*). To accommodate this flexibility, Co-Gym imposes minimal restrictions on agent implementation. To streamline agent integration, Co-

```
1  class EnvNode:
2      async def event_handler(self, channel, message):
3          if channel == "step":
4              action = message["action"]
5              role = message["role"]
6              private = False
7              self.update_last_step_timestamp()
8              // process action
9              if is_send_teammate_message(action):
10                 self.update_chat_history(action)
11             elif is_wait_teammate_continue(action):
12                 return
13             else:
14                 obs, reward, done, private = self.env.step(role, action)
15                 if done:
16                     yield "end", {...}
17                     ... // Clean up
18                     return
19             // send notification
20             if private:
21                 payload = self.get_payload(role)
22                 yield f"{role}/obs", payload
23             else:
24                 for role in self.team_members:
25                     payload = self.get_payload(role)
26                     yield f"{role}/obs", payload
27         elif channel == "tick":
28             if not self.exceed_idle_time():
29                 return
30             for role in self.team_members:
31                 payload = self.get_payload(role)
32                 yield f"{role}/obs", payload
```

Listing 1: Pseudo code of the Co-Gym notification protocol.

Gym includes an `AgentNode`, which wraps the LM agent to subscribe to notifications on the `{role}/obs` channel, adhering to the notification protocol in Listing 1. When a new message appears on the `{role}/obs` channel, the LM agent is responsible for deciding whether to take action and, if so, specifying the action. The `AgentNode` then sends the action string and agent role to the environment via the `step` channel, where the `EnvNode` processes it.

## C.2  DETAILS OF SUPPORTED TASK ENVIRONMENTS

As discussed in §4.1, we include three representative task environments as the initial set of benchmark problems to evaluate collaborative agents: Travel Planning, Related Work, and Tabular Analysis. In this section, we document the implementation details of these task environments.

**Travel Planning**  We leverage the medium and hard cases from the validation set of the TravelPlanner benchmark (Xie et al., 2024a) and use its database records to simulate search actions. Hard constraints for each task instance (*e.g.*, budget limits, special accommodation requests) are set as hidden information only visible to the simulated user. While the original TravelPlanner benchmark employs a binary final pass rate to measure success, we use its evaluation script to compute the task performance as the average of the commonsense pass rate and the constraint pass rate, providing a continuous measure of performance. In Co-Gym (Simulated), the search functions in the Travel Planning action space operate on databases constructed by Xie et al. (2024a). In Co-Gym (Real), real-time Google Search and Google Places APIs are utilized.

**Related Work**  We leverage the Computer Science (CS) category of the arXiv repository by selecting high-quality conference papers in various areas to construct initial queries. For each query, the hidden information for simulated humans is curated by extracting 3-9 hints such as required subheadings, citations, subsection counts, and writing style characteristics. For `SearchPaper` in Co-Gym (Simulated), we index papers from the arXiv CS category published prior to October 2024 using the `voyage-3` text embedding model and retrieve top 10 papers for each search query. For task-specific scoring function, we develop a 1-5 scoring rubric (see Figure 13) and use `gpt-4-06-13` as the judge model operating at a temperature of 0. Automated scores align with

human evaluations, with correlation coefficients of 0.791 (Pearson) and 0.741 (Spearman) across 20 sampled sections. In Co-Gym (Real), we use the arXiv search API[1] for `SearchPaper`.

**Tabular Analysis**  We use DiscoveryBench, a dataset designed for systems to derive hypotheses based on queries and provided tables (Majumder et al., 2024). We focus on instances from DiscoveryBench-Real that include unprocessed table or more than one table, which are considered challenging cases within the original benchmark. The domain knowledge and dataset metadata fields in the original dataset are treated as additional information available to the simulated human. Task performance is evaluated by assessing whether the derived hypothesis and the gold hypothesis are entailed using its evaluation script.

### C.3  CO-GYM (SIMULATED) USER SIMULATOR

In Co-Gym (Simulated), we employ an LM (`gpt-4o` in our experiments) to simulate human behavior. Since the simulated human also needs to perceive change programmatically, we implement the `SimulatedHumanNode`, which listens to notifications in the same way as the `AgentNode` (see Appendix C.1). Within the `SimulatedHumanNode` event loop, the simulated human processes observations and selects actions from five predefined action types that represent potential human behaviors:

- ANSWER QUESTION: The simulator LM further generates the answer and the next action would be `SendTeammateMessage`.

- PROVIDE FEEDBACK: The simulator LM further generates the feedback and the next action would be `SendTeammateMessage`, simulating human proactive information sharing.

- TAKE TASK ACTION: The simulator LM further generates an action string within the task-specific action space, simulating human task engagement.

- DO NOTHING: The next action would be `WaitTeammateContinue`, simulating the human tendency to pause, reflect or expect the agent to do the actual work.

- FINISH: Notify the environment to end the task.

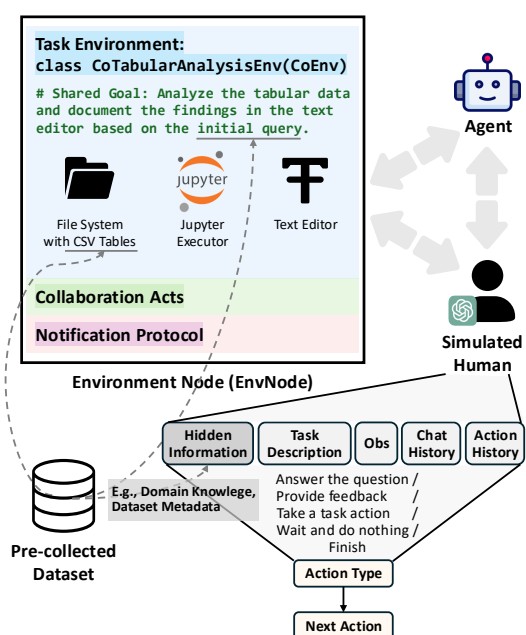

Figure 3: Illustration of Co-Gym (Simulated). The human is simulated by a language model using hidden information associated with each task case. The hidden information is not visible to the LM agent.

To create the potential dynamics within the human-agent team, where the human may have additional knowledge, preferences, or insights about the task, we curate hidden information for each task instance which is only visible to the simulator LM. As illustrated in Figure 3, the simulator chooses the action type and obtains the next action based on such hidden information together with the task description, current observation, chat history, and action history.

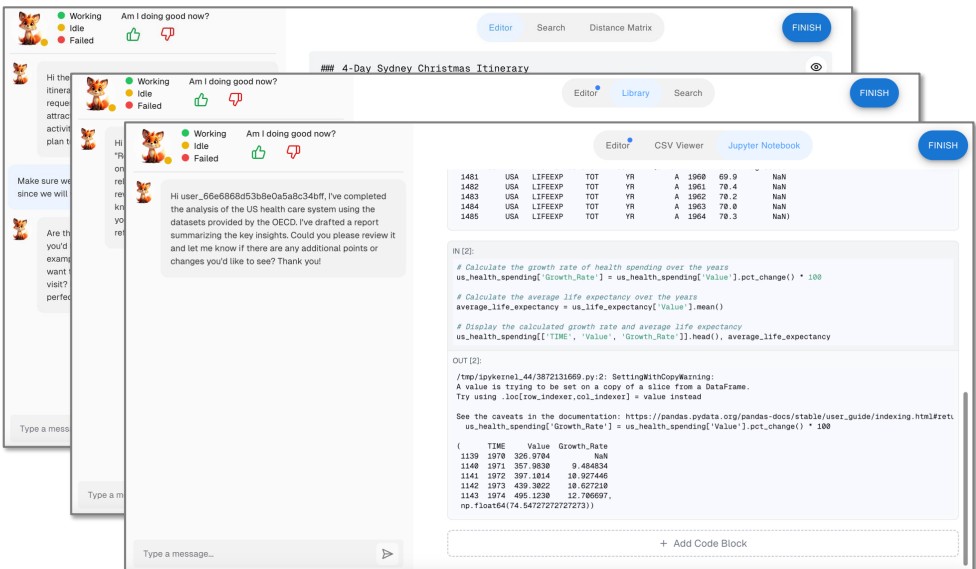

Figure 4: Screenshots of the interactive web application for Co-Gym (Real).

### C.4  CO-GYM (REAL) USER INTERFACE

To enable real-time collaboration between human users and LM agents in the Co-Gym (Real) condition, we developed web applications tailored for each task. These applications feature a chat panel and a shared workspace. Due to the non-turn-taking interaction design of Co-Gym, human users can perform task actions or send messages to the LM agent at any time. The user interface provides visual signals to notify changes, adhering to the notification protocol. Figure 4 illustrates the web application. We host the web application on a CPU server with 4 cores and 16 GB of memory.

Once the task is completed, we ask the human participant to: (1) rate the final outcome on a scale of 1 to 5, (2) rate their Overall Satisfaction with the collaboration process on a scale of 1 to 5, (3) provide pairwise comparison rating of the outcome against the result obtained by a fully autonomous agent based on the initial query. Additional evaluation details are provided in Appendix F.

## D  USER SIMULATOR VALIDATION

**Quality Coding**  We sample 100 simulated trajectories each from three tasks and recruit annotators from Prolific to annotate the simulator quality in terms of accuracy, consistency, and plausibility.

Specifically, the annotation rubric is as follows:

- **Accuracy**: Does the user simulator respond appropriately based on its internal goal and hidden information?
- **Consistency**: Is the user simulator internally coherent across turns?
- **Plausibility**: Does the behavior resemble a reasonable human response?

Table 7: User simulator quality evaluation based on the percentage of positive ratings determined by majority vote.

| Accuracy | Consistency | Plausibility |
|----------|-------------|--------------|
| 93.0%    | 92.0%       | 90.3%        |

Each trajectory received three independent annotations. We report the percentage of positive ratings based on majority vote in Table 7.

**Comparison of Simulated and Real Trajectories**  We further conduct a pairwise trajectory annotation task. Annotators are shown two trajectories performing the same task and are informed

---

[1]https://github.com/lukasschwab/arxiv.py

that one or both trajectories may be simulated. For each pair, annotators are asked whether they can confidently identify which trajectory involves a simulated user. They are given three options: (1) the two trajectories are indistinguishable, (2) Trajectory A is simulated, or (3) Trajectory B is simulated. To control for bias and evaluate baseline distinguishability, we include 50 simulated–human pairs and 50 human–human pairs in the annotation pool. Each pair received three annotations.

Among the simulated–human pairs, 26% of the annotations judged the two trajectories to be indistinguishable, suggesting that the simulated users often behave similarly to real users. When annotators chose a specific trajectory as simulated (*i.e.*, distinguishable cases), their accuracy was 56.9%, which is not significantly above chance (binomial test, $p = 0.0898$). These findings indicate that our simulated users are often difficult to distinguish from real users, supporting the realism and plausibility of our simulator.

# E IMPLEMENTATION DETAILS OF LM AGENTS

As described in §5.1, we implement LM agents with ReAct-style prompting. We further incorporate a Scratchpad module for in-session memory as trajectories are usually long in the three tasks we study, especially in a human-agent collaboration setup. This section details the implementation of the Collaborative Agent with Situational Planning, the best-performing method in our experiments, which employs a two-stage decision-making process to handle notifications. Figure 5 illustrates the agent workflow when processing a notification received via the event loop in `AgentNode`.

**System Prompt**  The system prompt describes the current task and emphasizes that the agent shall collaborate with its teammate(s) to complete the task. It includes all relevant information at the current timestamp, such as the latest scratchpad state, observations, and chat history between the human and the agent, as depicted in Figure 6.

**Scratchpad Update** (Figure 5 ①)  A key challenge in human-agent collaboration is enabling the agent to dynamically replan in response to new requests or suggestions while maintaining task progress. Also, human involvement often results in longer trajectories compared to fully autonomous agents. To enable the agent to record important information for future use during the session, we implement the in-session memory (*i.e.*, Scratchpad) as a dictionary, allowing for the insertion, deletion, and editing of items. When a new notification arrives, the system prompt is concatenated with the scratchpad update prompt (Figure 7), instructing the LM to update the scratchpad based on the current observation and chat history.

**Situational Planning** (Figure 5 ②)  Collaborative agents face a more complex action space than fully autonomous agents, as they must coordinate and communicate with humans to keep them informed about critical decisions, elicit latent preferences, and leverage their expertise. Unlike the baseline Collaborative Agent, which performs poorly in Co-Gym (Simulated) results (see Table 3), the Collaborative Agent with Situational Planning incorporates a three-way classification prompt instead of directly generating the action string. This prompt guides the agent to decide whether to take a task action, communicate with teammate(s), or wait for teammate(s) to proceed. The classification is achieved through in-context few-shot learning, with the prompt template shown in Figure 8. This additional step encourages explicit consideration of coordination and communication, resulting in more balanced initiative-taking, proactive confirmations, and improved task performance.

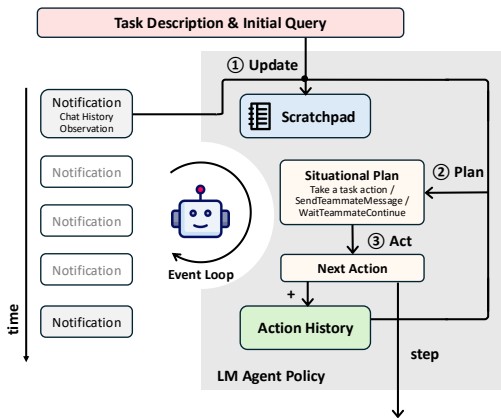

Figure 5: The workflow of Collaborative Agent with Situational Planning to process a notification received by the event loop in `AgentNode`.

**Action Taking** (Figure 5 ③) After determining the plan, the agent generates an action string depending on the decision: If the plan is to take a task action, the LM is prompted to generate the corresponding action string, similar to a Fully Autonomous Agent; if the plan is to communicate with teammate(s), the LM is prompted to generate a message and we construct the action string accordingly; if the plan is to wait, the `AgentNode` remains idle and listens for upcoming notifications. The action string, when constructed, is sent to `EnvNode` via the `step` channel.

---

**System Prompt**

SETTING: Your name is {name}. You are a helpful AI Agent who can take actions to interact with the environment and collaborate with other team members (e.g., the user) to complete the task. Your goal is to complete the task and aim for a high task performance rating.

You need to collaborate with your teammates effectively because they may have additional expertise or have preference/information important to the task. There are the following members in the team: {team_members}.

TASK DESCRIPTION:
{task_description}

SCRATCHPAD:
Here is the scratchpad that you use to take notes or store information in previous steps, which serves as your memory:
{scratchpad}

OBSERVATION:
Here is the current observation that reveals the current status of the task environment:
{observation}

COMMUNICATION:
Here is the current chat history that records the messages exchanged between you and other teammates (e.g., the user):
{chat_history}

---

Figure 6: The system prompt for Collaborative Agent and Collaborative Agent with Situational Planning. The system prompt will be populated with information provided in the notification.

**Scratchpad Updating Prompt**

Note that the current environment observation may change when you or your teammate(s) take actions. Remember to update your scratchpad accordingly if needed.
Guidelines:
1. Keep your scratchpad concise and relevant.
2. If there is any information that could be useful for future steps but not in the scratchpad, add it to the scratchpad.
3. If every information in the current observation is already in the scratchpad, you do not need to update the scratchpad.
4. If a past action does not lead to any progress, consider updating the scratchpad to remind yourself of not repeating the same action.

ACTION SPACE SPECIFICATION:
You can choose from and only from the following actions to manipulate your scratchpad. You can only choose one action at a time. Invalid actions will hurt your performance rating.
The following actions are available:
Add a note to the scratchpad (Parameters: ['note_id', 'note'])
- Description: Add a note to the scratchpad with the provided note_id and note.
- Regex pattern for the action (your output needs to follow this if you take this action): 'ADD_NOTE(note_id=(.*), note=(.*))'

Delete a note from the scratchpad (Parameters: ['note_id'])
- Description: Delete a note from the scratchpad with the provided note_id.
- Regex pattern for the action (your output needs to follow this if you take this action): 'DELETE_NOTE(note_id=(.*))'
Edit a note in the scratchpad (Parameters: ['note_id', 'note'])
- Description: Edit a note in the scratchpad with the provided note_id.
- Regex pattern for the action (your output needs to follow this if you take this action): 'EDIT_NOTE(note_id=(.*), note=(.*))'

Do nothing (Parameters: [])
- Description: Choose this action if there is no need to update the scratchpad. Do not spam the scratchpad with unnecessary updates.
- Regex pattern for the action (your output needs to follow this if you take this action): 'DO_NOTHING()'

OUTPUT FORMAT:
Give your output in the format of "Thought:...\nAction:... (must follow the regex pattern of the selected action)".

Figure 7: Prompt for updating the agent scratchpad that serves as the in-session memory.

---

**Situational Planning Prompt**

Now, based on the current situation, decide to either:
1. Send a message to your teammate(s) (e.g., ask a question, request feedback, etc.) to facilitate collaboration.
2. Take a task action to change the task environment observation.
3. Do nothing to allow your teammate(s) to take actions.

To ensure your are collaborating effectively, remember to:
1. Communicate clearly and effectively with your teammate(s) (e.g., the user).
2. Wait for other teammates to respond if your previous action requires a response. Do not spam the chat.
3. Coordinate and synchronize your actions with the user or other teammates.
4. Help establish task and role expectations with your teammates if you need their expertise.
5. Take your teammates' cognitive load into consideration when making decisions. You should not ask them to debug your own code or ask too many questions at the same time.

OUTPUT FORMAT:
Give your output in the format of "Thought:...\nPlan: 1. Send a message / 2. Take a task action / 3. Do nothing".
——
Example 1:
TASK DESCRIPTION:
The task is to analyze the user-provided tabular data to identify patterns and insights.
SCRATCHPAD: ...
OBSERVATION:
jupyter_history:
Code block:
```
import pandas as pd
df = pd.read_csv('data.csv')
print(df.columns)
```
Output:
```
Index(['Age', 'Education', 'Occupation', 'Computer Usage', 'Experience with LLM', 'LLM
Usage Frequency', 'Workflow 1', 'Workflow 2', 'Time 1', 'Preferred Assis 1', 'Time 2',
'Preferred Assis 2'], dtype='object')
```
COMMUNICATION: No chat history.
ACTION HISTORY: ...

Thought: I need to understand the data better before taking any action. There is no additional information about each column and the background of the data. I need to send a message to the user to ask for more information. Plan: 1. Send a message
——
Example 2:
TASK DESCRIPTION:
The task is to plan a trip to Paris.
SCRATCHPAD: ...
OBSERVATION: ...
COMMUNICATION:
You: Could you please provide me with your preferred travel dates? Are there any specific places you would like to visit in Paris?
ACTION HISTORY: ...

Thought: I have asked the user for their preferred travel dates and places to visit in Paris. I need to wait for the user's response before taking any further action. Plan: 3. Do nothing
——
Example 3:
TASK DESCRIPTION:
The task is to write a related work section for my paper around Human-AI collaboration.
SCRATCHPAD: ...
OBSERVATION: ...
COMMUNICATION: ...
ACTION HISTORY: No actions taken yet.

Thought: I need to start by reviewing the existing literature and papers related to Human-AI collaboration. I should take a task action to search for relevant papers. Plan: 2. Take a task action

---

Figure 8: Prompt for situational planning that classifies the current context into one of three action categories: taking a task action, sending a message, or waiting for teammate(s) to proceed.

Table 8: Percentage of trajectories hitting agent action number limit (*i.e.*, 30).

|  |  | Travel Planning | Related Work | Tabular Analysis |
|---|---|---|---|---|
| Fully Autonomous Agent | GPT-4o | 0.01 | 0.03 | 0.00 |
|  | GPT-4-turbo | 0.00 | 0.01 | 0.00 |
|  | Claude-3.5-sonnet | 0.00 | 0.00 | 0.00 |
|  | Llama-3.1-70B | 0.34 | 0.25 | 0.53 |
| Collaborative Agent | GPT-4o | 0.01 | 0.12 | 0.00 |
|  | GPT-4-turbo | 0.00 | 0.04 | 0.00 |
|  | Claude-3.5-sonnet | 0.11 | 0.04 | 0.00 |
|  | Llama-3.1-70B | 0.60 | 0.18 | 0.18 |
| Collaborative Agent with Situational Planning | GPT-4o | 0.00 | 0.25 | 0.00 |
|  | GPT-4-turbo | 0.00 | 0.09 | 0.00 |
|  | Claude-3.5-sonnet | 0.25 | 0.02 | 0.02 |
|  | Llama-3.1-70B | 0.42 | 0.21 | 0.19 |

## F    EVALUATION DETAILS

§3.3 outlines the evaluation suite that assesses collaborative agents along two dimensions: outcome and process. Below, we provide more details on the computation of these metrics.

### F.1    METRICS FOR COLLABORATION OUTCOME

**Delivery Rate**    Agents are given a step limit of 30 in our experiments. A task is considered "delivered" if the editor is not empty upon completion of the task or when the step limit is reached. We set the step limit of 30 following previous work (Xie et al., 2024a). As shown in Table 8, most agents do not hit the step limit except for agents powered by Llama-3.1-70B.

**Task Performance**    Each task environment in Co-Gym specifies a task-specific scoring function to quantify Task Performance (§3.3). This function may be a deterministic metric or based on LM/human judgments. Please refer to §4.2 for details.

### F.2    METRICS FOR COLLABORATION PROCESS

**Initiative Entropy** ($H_{\textbf{init}}$)    For each message in the chat history, we use Llama-3.1-70B to determine whether it demonstrates initiative-taking. The prompt used for this is shown in Figure 12. The prompt is based on the framework proposed by Chu-Carroll & Brown (1997), which defines initiative in collaborative dialogues as either directing task execution ("task initiative") or facilitating mutual understanding ("dialogue initiative"). Using these judgments, $H_{\text{init}}$ is computed according to Equation (1). We validate the prompt by randomly sampling 60 cases and having two human annotators provide judgments. The average Cohen's $\kappa$ between Llama-3.1-70B and the annotators was 0.425 and the joint Fleiss' $\kappa$ score was 0.310. These results indicate moderate agreement between model predictions and human annotations (Mukaka, 2012). To aid interpretation, we also include a walk-through of how $H_{\text{init}}$ is computed and present example trajectories at different $H_{\text{init}}$ levels in Figure 9, Figure 10, Figure 11.

Table 9: Scoring rubric for Overall Satisfaction in Co-Gym (Real).

| Score 1 Description | **Extremely dissatisfied:** The agent communicates poorly all the time and is not helpful for the task at all. |
|---|---|
| Score 2 Description | **Somewhat dissatisfied:** The agent communicates poorly and is not very helpful for the task. |
| Score 3 Description | **Neutral:** The agent can have meaningful communication and is somewhat helpful for the task. |
| Score 4 Description | **Somewhat satisfied:** The agent communicates effectively overall and is helpful for the task. |
| Score 5 Description | **Extremely satisfied:** The agent communicates effectively all the time and is very helpful for the task. |

**Overall Satisfaction**    We compute this metric in Co-Gym (Real) condition by requesting the human to rate their overall satisfaction with the agent on a 1-5 scale according to the description in Table 9.

---

**High-$H_{\text{init}}$ Example: Both human and agent actively take initiative**

*Agent takes 5 InternetSearch actions, 4 BusinessSearch actions; User takes 2 InternetSearch actions*
*Agent takes 1 EditorUpdate action*
**[User] SendTeammateMessage** make sure the flight is usiness class and offers a lay flat bed
    (*Judged as taking initiative: directs task execution*)
*Agent takes 1 EditorUpdate action*
**[Agent] SendTeammateMessage** Hi! I've put together a draft itinerary for your one-week trip to London. It includes business class flights with lay-flat beds, a stay at The Waldorf Hilton, and a selection of attractions and restaurants. Please let me know if there's anything you'd like to adjust or add to the plan.
    (*Taking initiative: establishes mutual belief*)
**[Agent] SendTeammateMessage** Hi! I've drafted your travel itinerary, including business class flights with lay-flat beds, a stay at The Waldorf Hilton, and a selection of attractions and restaurants in London. Please let me know if there's anything you'd like to adjust or add to the plan. Your feedback is important to ensure the itinerary meets your expectations!
    (*Taking initiative: establishes mutual belief + solicits suggestions*)
**[User] SendTeammateMessage** oh ya, for the travel plan. i mean it takes 7 days include departure and returning time
    (*Not taking initiative: answers the agent's question*)
*Agent takes 4 EditorUpdate actions*
**[User] SendTeammateMessage** wht the best time to visit british museum
    (*Taking initiative: drives task execution*)
**[Agent] SendTeammateMessage** The best time to visit the British Museum is during weekdays, especially in the early morning or late afternoon, to avoid crowds. Let me know if you'd like to adjust the itinerary to fit this timing!
    (*Taking initiative: establishes mutual belief + proposes next step*)
**[User] SendTeammateMessage** is the tower of london best landmark on london, i mean best one must visit
    (*Taking initiative: drives task execution*)
**[Agent] SendTeammateMessage** The Tower of London is indeed one of the most iconic and historically significant landmarks in London, making it a must-visit attraction. Would you like to prioritize it as a must-see on your itinerary, or are there other attractions you'd prefer to include as well? Let me know your preferences so I can tailor the itinerary to your interests!
    (*Taking initiative: establishes mutual belief + proposes next step*)
**[User] SendTeammateMessage** can you show me the options for the flight, means like in what times, or is it transit or direct flight
    (*Taking initiative: drives task execution*)
*Agent takes 7 InternetSearch actions, 2 EditorUpdate actions; User takes 1 DistanceMatrix action, 4 InternetSearch actions*
**[User] SendTeammateMessage** can you make it clear or just simply make an options for day6
    (*Taking initiative*)
**[Agent] SendTeammateMessage** Here are some flight options ... For Day 6 you might consider these options: ...
    (*Not taking initiative: executes user request*)
**[User] SendTeammateMessage** can you put it on travel plan
    (*Taking initiative: drives task execution*)
*Agent takes 1 EditorUpdate; User takes 1 EditorUpdate and 5 DistanceMatrix actions*
**[Agent] SendTeammateMessage** I've updated the travel plan with the flight options and Day 6 itinerary choices. Please review the updated plan in the shared editor and let me know if there's anything else you'd like to adjust or add!
    (*Taking initiative: establishes mutual belief + proposes next step*)
*User takes 2 EditorUpdate actions and finishes task*

---

Figure 9: A human-agent collaboration trajectory with high $H_{\text{init}}$. In this example, $p_{\text{agent}} = 5/11$, $p_{\text{user}} = 6/11$, and $H_{\text{init}} = 0.994$. Typos in user messages are preserved from the original data.

---

**A lower $H_{\text{init}}$ case where the user primarily takes initiative.**

*Agent takes 1 SearchArxiv action, 1 AddPaperToLibrary action, 1 LibraryToDraft action*

**[Agent] `SendTeammateMessage`** Hi user_xxx! I've collected several research papers related to data leakage and memorization in language model benchmarks. Do you have any specific preferences or additional information that you think should be included in the "Related Works" section? Let me know if there's anything specific you'd like me to focus on or any other details you can provide.

    (*Taking initiative: establishes mutual belief + proposes next step*)

**[User] `SendTeammateMessage`** do you have the capability to talk about the [1] [2] or [3] sources that are in the library?

    (*Taking initiative: drives task execution*)

**[Agent] `SendTeammateMessage`** Sure! Here's a brief overview of the sources:

- [1] *Quantifying and Analyzing Entity-level Memorization in Large Language Models*: introduces a method for quantifying memorization at the entity level in LLMs, highlighting privacy risks and the ability of models to reproduce training data even with partial leakages.

- [2] *Are Large Language Models Memorizing Bug Benchmarks?*: investigates data leakage in bug benchmarks used for evaluating LLMs in software engineering tasks, showing how memorization can inflate performance metrics.

- [3] *Inference-Time Decontamination: Reusing Leaked Benchmarks for Large Language Model Evaluation*: presents Inference-Time Decontamination (ITD) for detecting and rewriting leaked samples to provide more accurate performance evaluation.

    (*Not taking initiative: answers the user's question*)

**[User] `SendTeammateMessage`** what are some papers on how bad memorization has gotten?

    (*Taking initiative: drives task execution*)

*Agent takes 2 SearchArxiv actions, 1 EditorUpdate action*

**[User] `SendTeammateMessage`** can you give me a more in-depth summary of this bug-benchmarks paper?

    (*Taking initiative: drives task execution*)

**[Agent] `SendTeammateMessage`** Certainly! Here's a more in-depth summary of the *Are Large Language Models Memorizing Bug Benchmarks?* paper: ...

    (*Not taking initiative: answers the user's question*)

Figure 10: A human-agent collaboration trajectory with low $H_{\text{init}}$. In this example, $p_{\text{agent}} = 1/4$, $p_{\text{user}} = 3/4$, and $H_{\text{init}} = 0.811$.

---

**A very low $H_{\text{init}}$ case where it's mainly the agent that takes initiative.**

*Agent takes 3 ExecuteJupyterCell actions; User takes 1 EditorUpdate action*

**[Agent] `SendTeammateMessage`** Hi user_xxx! As I begin analyzing the impact of COVID-19 on birth rates, do you have any specific variables or aspects you would like me to focus on? This could include particular birth outcomes, demographic factors, or any other elements you find important. Let me know so I can tailor the analysis to your needs!
  (*Taking initiative: establishes mutual belief*)

**[User] `SendTeammateMessage`** Demographic factors , other factors and time frame
  (*Not taking initiative: answers the agent's question*)

**[Agent] `SendTeammateMessage`** Thank you for sharing your preferences! Could you please specify which demographic factors or time frames you are particularly interested in? This will help me tailor the analysis to better meet your needs.
  (*Not taking initiative: repeats the previous question*)

**[User] `SendTeammateMessage`** Age, income level, urban vs rural     (*Not taking initiative: answers the agent's question*)

**[Agent] `SendTeammateMessage`** Could you please specify any particular time frames you are interested in for analyzing the impact of COVID-19 on birth rates? This will help me focus the analysis on the relevant data.
  (*Taking initiative: establishes mutual belief*)

**[User] `SendTeammateMessage`** data from 2020 to 2022
  (*Not taking initiative: answers the agent's question*)

*Agent takes 1 ExecuteJupyterCell action*

**[Agent] `SendTeammateMessage`** Hi user_xxx! I've encountered an issue while cleaning the COVID-19 pregnancy dataset. There are non-numeric values in the `'Total1'` column, such as `'Number'`, which are causing errors during conversion. Do you have any specific instructions on how you would like me to handle these non-numeric values? For example, should I remove these rows, replace them with a specific value, or take another approach? Let me know how you'd like to proceed!
  (*Taking initiative: establishes mutual belief + asks for help at a critical step*)

**[User] `SendTeammateMessage`** remove rows with non numeric values
Let me know how you'd like to proceed!
  (*Not taking initiative: answers the agent's question*)

**[Agent] `SendTeammateMessage`** Thank you for your guidance! I will proceed with removing rows that contain non-numeric values in the COVID-19 pregnancy dataset. Once the data is cleaned, I'll continue with the analysis to assess the impact of COVID-19 on birth rates and the correlations between birth outcomes and COVID-19 incidence among pregnant women. I'll keep you updated on the progress!
  (*Taking initiative: establishes mutual belief*)

*Agent takes 2 ExecuteJupyterCell actions*

**[User] `SendTeammateMessage`** Okay than you
  (*Not taking initiative*)

*Agent takes 8 ExecuteJupyterCell actions*

**[Agent] `SendTeammateMessage`** Hi user_xxx! As I continue with the analysis, could you please specify any particular birth outcomes you would like me to focus on? This could include factors like birth weight, preterm births, or any other outcomes you find important. Also, I've been working on cleaning the datasets and analyzing the monthly birth statistics. If you have any additional preferences or questions, feel free to let me know!
  (*Taking initiative: establishes mutual belief*)

**[User] `SendTeammateMessage`** access to prenatal care, fetal distress, gestational age. birth weight
  (*Not taking initiative: answers the agent's question*)

*Agent takes 4 ExecuteJupyterCell actions, 1 EditorUpdate action; User finishes the task*

---

Figure 11: A human–agent collaboration trajectory with very low $H_{\text{init}}$. In this example, $p_{\text{agent}} = 5/5$, $p_{\text{user}} = 0/5$, and $H_{\text{init}} = 0$. Grammar errors in user messages are preserved from the original data.

**Judging Initiative**

I am analyzing team member initiative in collaboration. Two types of utterance count as taking initiative:

1. Task Initiative: A team member is said to have the task initiative if she is directing how other member(s)' task should be accomplished, i.e., if her utterances directly propose actions that other members should perform.
- Examples: "Let's send engine E2 to Corning.", "Let's look at the first problem first.", "Let's consider driving from Fort Lauderdale to Louisiana and explore three cities there."
- Passive utterances like "Any suggestions", "Right, okay." are not considered as task initiative.

2. Dialogue Initiative: A team member is said to have the dialogue initiative if she tries to establish mutual beliefs. Both giving concrete information and asking concrete questions are considered dialogue initiative.
- Examples: "We can't go by Dansville because we've got Engine 1 going on that track.", "Would you like to consider traveling on a different date?", "What do you think about the first problem?"
- Repeating what the other person said, asking for clarification are not considered dialogue initiative.
Now given an utterance in the conversation, you need to judge whether the utterance takes initiative or not. Indicate your judgement with "Yes" or "No".
—
Utterance: {utterance}
Reasoning: Let's think step by step in order to

Figure 12: Prompt for judging initiative taking.

**Scoring Rubric**

You are an ACCURATE, FAITHFUL, CRITICAL, and FAIR judge. You will be given a related works section draft and a topic for the paper the related works were written for.

**Your Task:**

- Critically evaluate the quality of the related works section based on the Evaluation Criteria below.
- The final score MUST be an integer.
- Related Works sections that exhibit low quality attributes according to the evaluation criteria below must be given low overall scores.
- Related Works sections that exhibit high quality attributes according to the evaluation criteria below must be given high overall scores.
- Output the integer final score in your last sentence in the following format: "Therefore, the final score is..."
- Keep this rubric open while evaluating and reference the evaluation criteria below when assigning a score.
- You MUST give your final score based on the Evaluation Criteria given below.

Evaluation Criteria:
————————————————————————
**Score = 1:**
- Citation Usage: Only zero to four unique citations
- Relevance and Coverage: Content is off-topic or lacks any meaningful discussion of related works.
- Organization and Structure: Disorganized with no clear structure; ideas are scattered and hard to follow. Uses bullets points or list format in at least one subsection.
- Writing Style: Informal language; numerous grammatical errors; may use bullet points instead of prose.
- **Important**: If a prose style of writing is not present throughout the full related works (i.e., a subsection or subsections is in bullet point format) or if only 0 - 4 unique citations are present, an overall score of 1 should automatically be assigned regardless of the rest of the criteria.
————————————————————————
**Score = 2:**
- Citation Usage: ONLY five to six unique citations. Limited citations with minimal diversity; overuse of certain citations; inconsistent formatting.
- Relevance and Coverage: Touches on relevant topics superficially; lacks depth; may include irrelevant information. Most subsections only discuss one unique work and are not fleshed out.
- Organization and Structure: Some attempt at organization, but lack coherence; grouping of ideas is unclear. Subsections are put out of order or talks about current paper's work.
- Writing Style: Inconsistent academic tone; several grammatical errors; language may be unclear at times. Terminology like "In summary" are used.
————————————————————————
**Score = 3:**
- Citation Usage: Seven or more citations with some diversity; occasional over-reliance on certain sources; formatting is mostly consistent. If not all citations between the lowest cited index and the highest cited index appear at least once, an overall score of 3 is the highest that can be awarded.
- Relevance and Coverage: Addresses relevant topics adequately but lacks depth; discussions are generally accurate but not insightful.
- Organization and Structure: Logical structure is present; grouping of ideas is mostly coherent; transitions may need improvement.
- Writing Style: Maintains an academic tone with minor lapses; few grammatical errors; language is generally clear.
————————————————————————
**Score = 4:**
- Citation Usage: Multiple unique citations with good diversity; sources are well-distributed and support the text effectively; consistent formatting.
- Relevance and Coverage: Provides a comprehensive overview of relevant topics; discussions show understanding and some critical analysis.
- Organization and Structure: Well-organized with clear thematic grouping; ideas flow logically and coherently. Each subsection discussed multiple papers and has ideas and similarities fleshed out.
- Writing Style: Polished academic writing; clear and concise; minimal grammatical errors.
————————————————————————
**Score = 5:**
- Citation Usage: Wide variety of unique citations with excellent diversity; citations enhance the discussion significantly; flawless formatting.
- Relevance and Coverage: Thoroughly covers all relevant aspects of the topic with depth and insight; demonstrates critical analysis and synthesis.
- Organization and Structure: Exemplary organization with clear, logical progression; seamless transitions between ideas. There are clear and separated paragraphs that each contain a theme.
- Writing Style: Exceptional academic writing; language is precise, clear, and engaging; no grammatical errors.
————————————————————————

**Paper Topic:**

{topic}

**Related Works:**

{related_works}

**Evaluation Form:**

Figure 13: Scoring rubric for Related Work task environment in Co-Gym (Simulated).

### F.3    HUMAN EVALUATION DETAIL

For the experiment condition with Co-Gym (Real), we recruited human participants with relevant expertise or practical needs to collaborate with Collaborative Agent with Situational Planning powered by `gpt-4o`. Specifically, we targeted participants who had current needs in travel planning, tabular data analysis, or writing related work sections/literature surveys. Recruitment was initially conducted through word-of-mouth. To broaden our participant pool, we recruited travel planners through Upwork for Travel Planning and participants with Python programming and data analysis experience through Prolific to work with the agent on writing analytical reports from tabular data. Participants were compensated at a rate of $8.00 per hour, and the study received approval from our institution's Institutional Review Board (IRB).

In the ablation study (see §5.3), each participant (1) collaborated with a Collaborative Agent with Situational Planning in turn-taking Co-Gym, then (2) completed a matched **non–turn-taking** session on the original Co-Gym. After each session, they rated Task Performance (final outcome on a 1–5 scale, normalized to $[0, 1]$) and Overall Satisfaction (1–5 Likert) using the same rubric described above. Finally, they indicated a preference between conditions with a brief rationale.

## G    ERROR ANALYSIS

We conducted a comprehensive error analysis on trajectories collected from Co-Gym by developing a failure mode annotation taxonomy and annotating 150 trajectories each from Co-Gym (Real) and Co-Gym (Simulated) conditions. While certain errors can be partially alleviated through the Co-Gym framework itself (*e.g.*, C.1 "*Agents process tasks without informing users, resulting in a lack of progress awareness*" can be mitigated by streaming the agent action to the user interface), our analysis intentionally focuses on evaluating LM agents so as to illuminate concrete directions for their improvement. Accordingly, we attribute each error type to gaps in the LM's inherent capabilities, deficiencies in the design of the agent scaffolding (*e.g.*, memory, additional tools), or both. Table 10 summarizes the results.

Table 10: **Breakdown of common failure modes of LM agents during human-agent collaboration.** Failure mode statistics are derived from authors annotating 150 trajectories each from Co-Gym (Real) and Co-Gym (Simulated) conditions. These failure modes focus on LM agents as a whole, encompassing both the underlying LM and the agent scaffolding (*e.g.*, memory, additional tools, *etc.*). Failure modes are traced to these two dimensions, with 🔆 indicating failures from the underlying LM and 🔧 representing issues arising from the agent scaffolding.

| Failure Mode Description | Real | Simulated | Error Source |
|---|---|---|---|
| **Communication (C)** Real: 65% Simulated: 80% | | | |
| **(C.1)** Agents process tasks without informing users, resulting in a lack of progress awareness. | 23% | 24% | 🔆 |
| **(C.2)** Agents do not confirm or communicate their actions before execution. | 29% | 46% | 🔆 |
| **(C.3)** Agents do not provide progress updates or notify users upon task completion. | 33% | 27% | 🔆 |
| **(C.4)** Absence of estimated completion times impedes collaborative efficiency. | 15% | 13% | 🔆 🔧 |
| **(C.5)** Agents provide inadequate summaries after executing actions. | 14% | 14% | 🔆 |
| **(C.6)** Agents do not proactively seek clarification when user input is ambiguous or insufficient. | 11% | 5% | 🔆 |
| **(C.7)** Agents miss implicit cues to initiate expected actions. | 12% | 5% | 🔆 |
| **Situational Awareness (SA)** Real: 40% Simulated: 47% | | | |
| **(SA.1)** Agents disregard session context, treating each request as an isolated task. | 11% | 12% | 🔆 |
| **(SA.2)** Repetitive queries arise from neglecting prior interactions and user feedback. | 13% | 26% | 🔆 🔧 |
| **(SA.3)** Agents fail to process multiple user messages cohesively. | 11% | 9% | 🔆 |
| **(SA.4)** Agents deviate from prior instructions during extended sessions. | 3% | 7% | 🔆 🔧 |
| **(SA.5)** Agents execute critical actions without obtaining prior confirmation. | 18% | 8% | 🔆 |
| **(SA.6)** Agents do not adhere to instructions as session duration increases. | 10% | 15% | 🔆 🔧 |
| **Planning (PL)** Real: 39% Simulated: 43% | | | |
| **(PL.1)** Agents acknowledge tasks but fail to execute them. | 10% | 8% | 🔆 🔧 |
| **(PL.2)** Lack of task planning results in repeated or omitted actions. | 27% | 33% | 🔆 🔧 |
| **(PL.3)** Agents cannot revert previous actions when errors are identified, lacking initiative to correct them. | 3% | 3% | 🔆 |
| **(PL.4)** Agents fail to choose the optimal way to modify the environment, leading to inefficient operations. | 1% | 3% | 🔧 |
| **(PL.5)** Agents lack proactive planning abilities and cannot infer subsequent steps without explicit guidance. | 3% | 0% | 🔆 |
| **(PL.6)** Multi-faceted requests receive incomplete or partial responses. | 7% | 3% | 🔆 |
| **Environment Awareness (EA)** Real: 28% Simulated: 13% | | | |
| **(EA.1)** Agents do not assess the feasibility of requests within the constraints of available tools and resources. | 13% | 3% | 🔆 |
| **(EA.2)** Agents propose actions that they are unable to perform within the environment. | 2% | 0% | 🔆 |
| **(EA.3)** Agents fail to identify when external data or tools are required to fulfill a request. | 15% | 10% | 🔆 |
| **(EA.4)** Agents hallucinate inaccurate information by not utilizing available tools and external data. | 8% | 3% | 🔆 |
| **Personalization (P)** Real: 16% Simulated: 11% | | | |
| **(P.1)** Agents rely on rigid templates that do not adapt to individual user needs. | 7% | 5% | 🔧 |
| **(P.2)** Agents pose broad questions lacking specificity and clarity. | 5% | 3% | 🔆 |
| **(P.3)** Agents do not incorporate user preferences into future interactions, limiting personalization. | 6% | 5% | 🔆 🔧 |

## H    REPRESENTATIVE HUMAN-AGENT COLLABORATION TRAJECTORIES

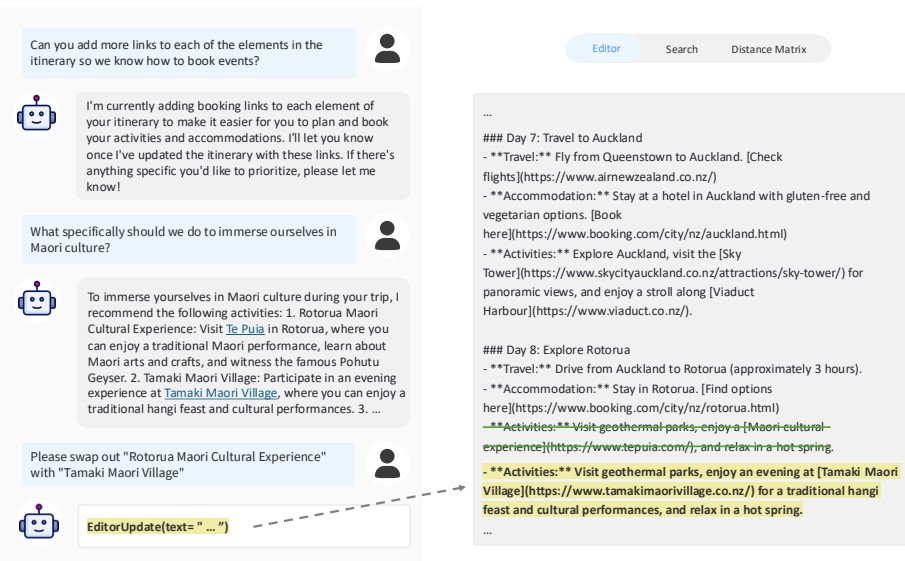

Figure 14: An example of effective human-agent collaboration in the Travel Planning task. The human seeks subjective recommendations from the agent on experiencing Maori culture. The agent offers relevant suggestions, defers the final decision to the human, and updates the plan accurately based on the human's choice.

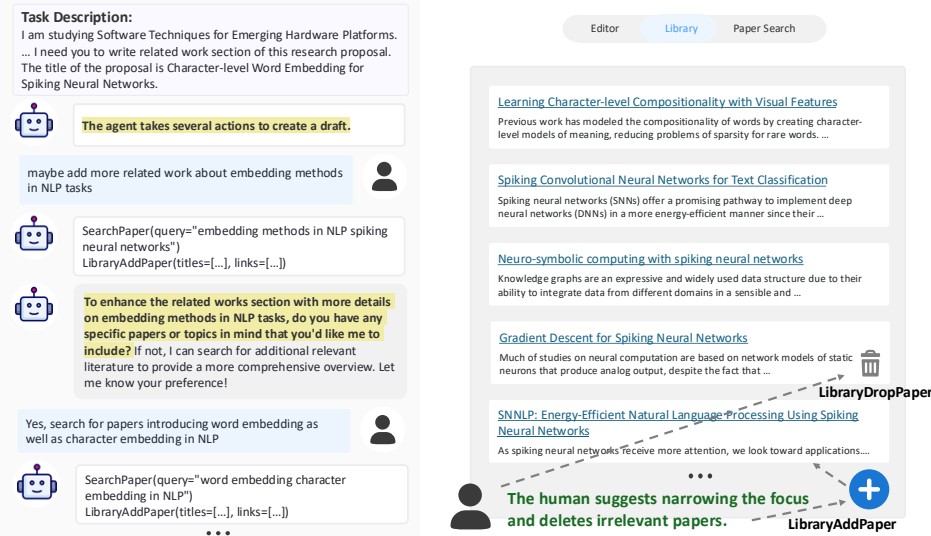

Figure 15: An example of effective human-agent collaboration in the Related Work task. The LM agent conducts search to generate the initial draft and asks clarifying questions based on the human's suggestions. The human reviews the collected papers and provides guidance leveraging their expertise.

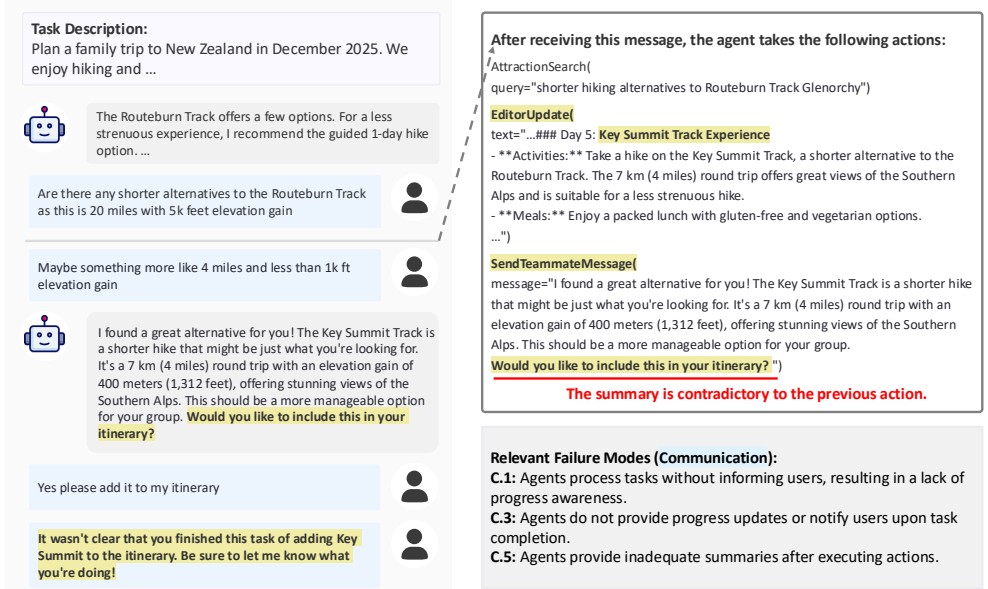

Figure 16: A partial example trajectory illustrating LM agent failures in the **Communication** category.

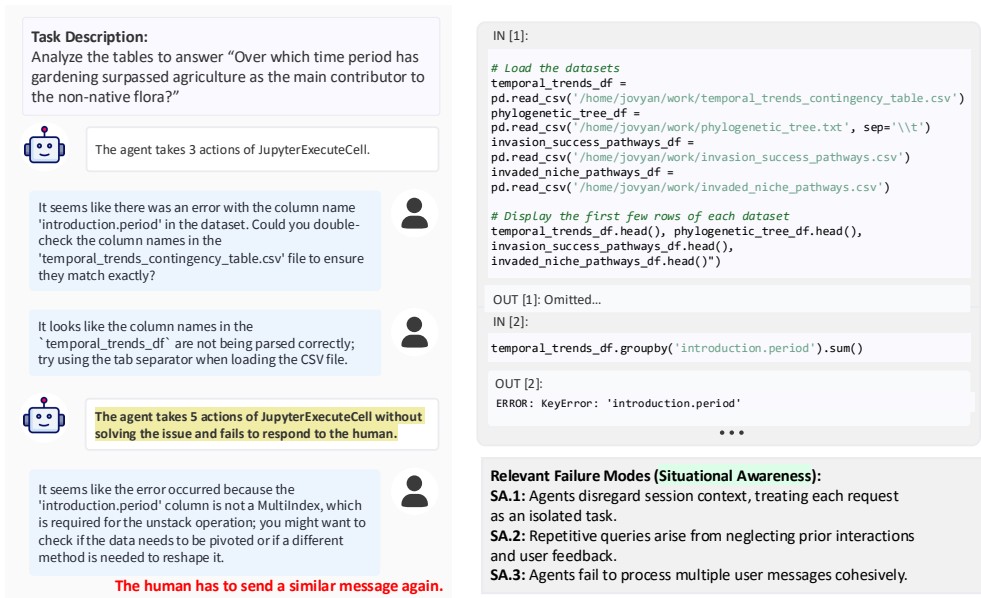

Figure 17: A partial example trajectory illustrating LM agent failures in the **Situational Awareness** category.

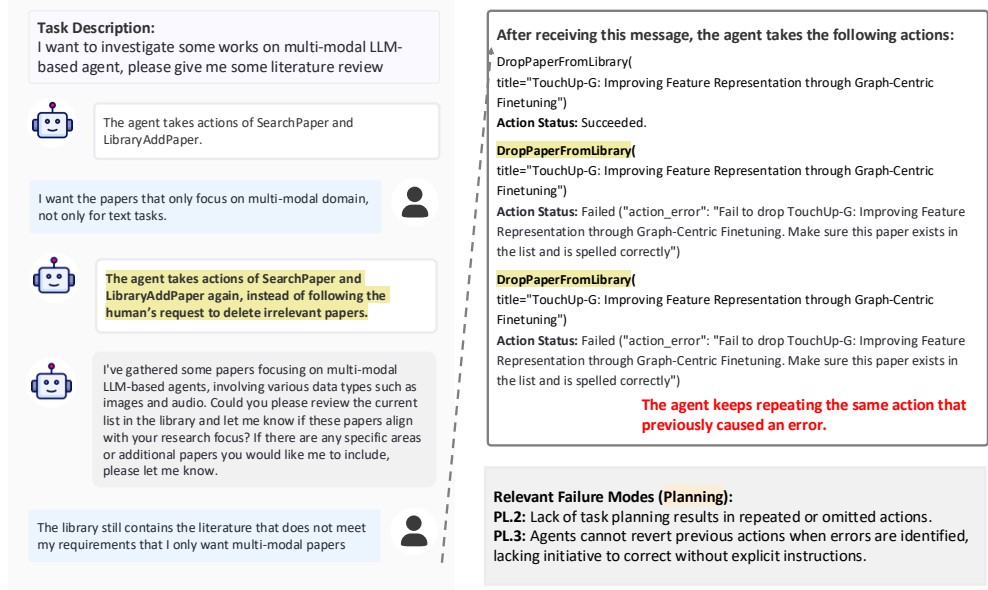

Figure 18: A partial example trajectory illustrating LM agent failures in the **Planning** category.

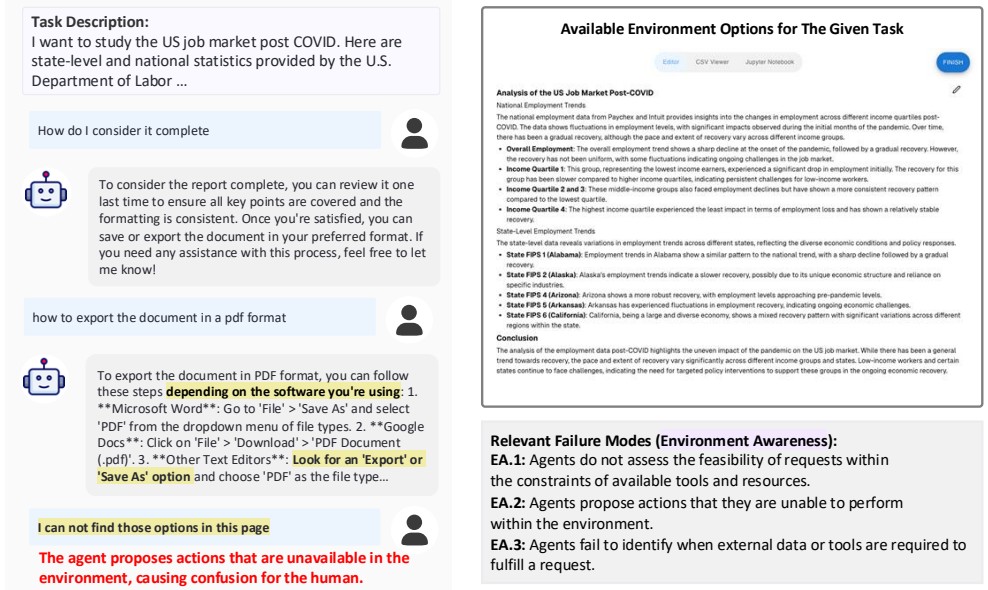

Figure 19: A partial example trajectory illustrating LM agent failures in the **Environment Awareness** category.

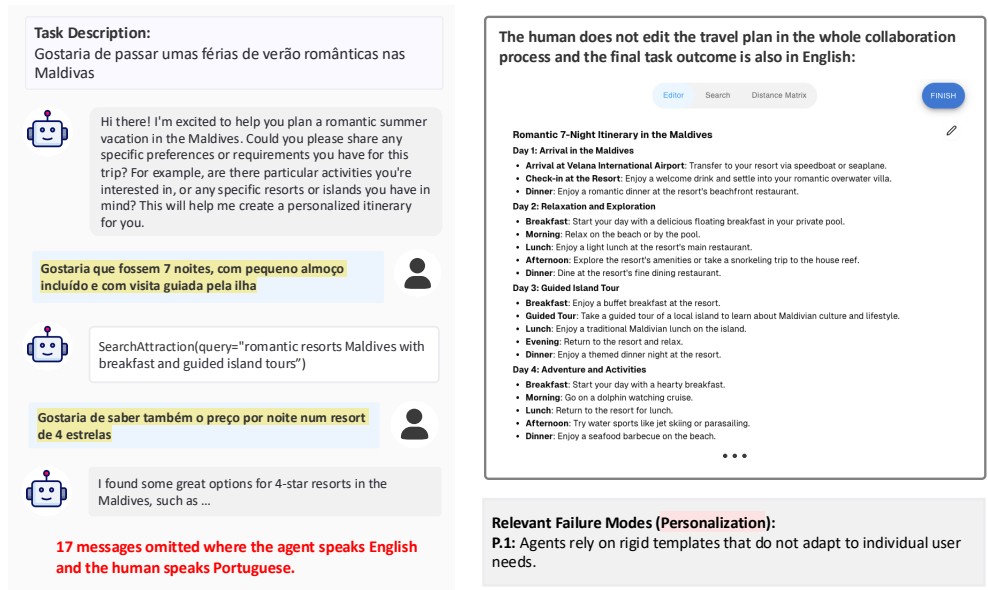

Figure 20: A partial example trajectory illustrating LM agent failures in the **Personalization** category.

